# *Drosophila melanogaster* Uncoupling Protein-4A (UCP4A) Catalyzes a Unidirectional Transport of Aspartate

**DOI:** 10.3390/ijms23031020

**Published:** 2022-01-18

**Authors:** Paola Lunetti, Ruggiero Gorgoglione, Rosita Curcio, Federica Marra, Antonella Pignataro, Angelo Vozza, Christopher L. Riley, Loredana Capobianco, Luigi Palmieri, Vincenza Dolce, Giuseppe Fiermonte

**Affiliations:** 1Department of Biosciences, Biotechnologies and Biopharmaceutics, University of Bari, 70125 Bari, Italy; paola.lunetti@unisalento.it (P.L.); rugorgo@gmail.com (R.G.); antonella.pignataro@uniba.it (A.P.); angelo.vozza@uniba.it (A.V.); luigi.palmieri@uniba.it (L.P.); 2Department of Biological and Environmental Sciences and Technologies, University of Salento, 73100 Lecce, Italy; 3Department of Pharmacy, Health and Nutritional Sciences, University of Calabria, Arcavacata di Rende, 87036 Cosenza, Italy; rosita.curcio@unical.it (R.C.); federica.marra@unical.it (F.M.); 4Department of Molecular Biosciences, College of Natural Sciences, The University of Texas at Austin, Austin, TX 78712, USA; chrisloganriley@gmail.com

**Keywords:** uncoupling proteins, *Dm*UCP4A, mitochondrial transporters, aspartate transport, ß-alanine, biogenic amines metabolism, Parkinson’s disease, CG6492, mitochondrial carrier family, N-acetylaspartate

## Abstract

Uncoupling proteins (UCPs) form a distinct subfamily of the mitochondrial carrier family (MCF) SLC25. Four UCPs, *Dm*UCP4A-C and *Dm*UCP5, have been identified in *Drosophila melanogaster* on the basis of their sequence homology with mammalian UCP4 and UCP5. In a Parkinson’s disease model, *Dm*UCP4A showed a protective role against mitochondrial dysfunction, by increasing mitochondrial membrane potential and ATP synthesis. To date, *Dm*UCP4A is still an orphan of a biochemical function, although its possible involvement in mitochondrial uncoupling has been ruled out. Here, we show that *Dm*UCP4A expressed in bacteria and reconstituted in phospholipid vesicles catalyzes a unidirectional transport of aspartate, which is saturable and inhibited by mercurials and other mitochondrial carrier inhibitors to various degrees. Swelling experiments carried out in yeast mitochondria have demonstrated that the unidirectional transport of aspartate catalyzed by *Dm*UCP4 is not proton-coupled. The biochemical function of *Dm*UCP4A has been further confirmed in a yeast cell model, in which growth has required an efflux of aspartate from mitochondria. Notably, *Dm*UCP4A is the first UCP4 homolog from any species to be biochemically characterized. In *Drosophila melanogaster*, *Dm*UCP4A could be involved in the transport of aspartate from mitochondria to the cytosol, in which it could be used for protein and nucleotide synthesis, as well as in the biosynthesis of ß-alanine and N-acetylaspartate, which play key roles in signal transmission in the central nervous system.

## 1. Introduction

Uncoupling proteins (UCPs) belong to the superfamily of mitochondrial carrier proteins (MCF) [1,2]. The term ‘uncoupling’ was originally coined to describe a condition in which mitochondrial oxygen consumption was not coupled to oxidative phosphorylation (i.e., ATP production). Mammalian brown adipocytes express a member of the MCF, named UCP1, which is able to uncouple oxidative phosphorylation by catalyzing a fatty acids-dependent proton leak across the inner mitochondrial membrane and has a key role in non-shivering thermogenesis. Since 1997, other members of the MCF were classified as putative uncouplers on the basis of their sequence homology with UCP1. The expression of UCP1 homologs in other tissues raised great expectations as potential targets for cold adaptation and enhanced energy expenditure, and even distant MCF members were named “uncoupling proteins”. Early studies on animal models and the discovery of UCP1 homologs in plants and ectothermic species ruled out any involvement of such proteins in non-shivering thermogenesis. More recently, a fatty acids-dependent proton transport activity catalyzed by UCP2 and UCP3, the two closest homologs of UCP1, was discredited [3]. Furthermore, some UCP1 homologs functionally reconstituted into artificial lipid vesicles acted as metabolite transporters: (i) human UCP2 catalyzed an exchange reaction of aspartate, malate, or oxaloacetate for phosphate plus a proton [4]; (ii) hUCP5 and hUCP6 transported sulfite, thiosulfate, and to lower extents dicarboxylates and aspartate [5]; (iii) plant UCP1 homologs were also functionally characterized as amino acids/dicarboxylates transporters [6].

Based on the UCP-specific sequence signatures located in the first, second, and fourth α-helices, four *Drosophila melanogaster* MCF members were classified as uncoupling proteins, *Dm*UCP4A, *Dm*UCP4B and *Dm*UCP4C that are homologous to human UCP4 (hUCP4), and *Dm*UCP5 that is homologous to hUCP5. *Dm*UCP4A-C and *Dm*UCP5 are encoded by the *CG6492, CG18340, CG9064*, and *CG7314* genes, respectively [1,7,8]. It should be emphasized that the four fruit fly UCP homologs share only 26–28% of protein identity with the archetypal uncoupler UCP1 [7], and the same UCP-specific sequence signatures were also found in the yeast mitochondrial dicarboxylate carrier [9].

In two *D. melanogaster* models of Parkinson’s disease (PD), *pink1* and *parkin* mutants, *Dm*UCP4A overexpression rescued mitochondria-specific phenotypes associated with mitochondrial membrane potential, by lowering reactive oxygen species production and increasing the resistance to oxidative stress [10]. The expression of *Dm*UCP4A in *pink1/parkin* mutants increased the efficiency of the electron transport chain, by increasing the oxygen consumption rate coupled with a higher ATP synthesis [10]. Furthermore, in both PD fruit fly models *Dm*UCP4A expression rescued locomotor defects, muscle degeneration, male sterility and spermatid morphological alterations, and prolonged dopaminergic neuron survival by protecting them from degeneration [10]. This last finding was also reported in the knockout of the PD-associated mitochondrial gene CHCHD2 overexpressing *Dm*UCP4A [11]. Da-Ré and co-workers demonstrated that, among the three *Dm*UCP4s, only *Dm*UCP4C had protonophoric activity, which was activated by palmitic acid and inhibited by GDP [12] and the heat production resulting from mitochondrial uncoupling was crucial for the survival of *D. melanogaster* larvae and their development at low temperatures [12]. *Dm*UCP4C uncoupling activity was also linked to *D. melanogaster* longevity [13]. The extended lifespan observed in *D. melanogaster* lacking the circadian clock components PERIOD and TIMELESS was associated with altered cellular respiration via a *Dm*UCP4C-dependent increased mitochondrial uncoupling, which could be further supported by *Dm*UCP4B [13]. However, Ulgherait and coworkers ruled out any possible involvement of *Dm*UCP4A in *D. melanogaster* longevity and mitochondria uncoupling [13]. A fatty acids-dependent protonophoric activity was also associated with *Dm*UCP5 expressed in yeast mitochondria [14], although the same authors found no difference in the rate of oxygen consumption and ATP production between wild type and *Dm*UCP5 knock-out flies [15].

Since none of the previously published works supported any *Dm*UCP4A-mediated uncoupling activity, and the *Caenorhabditis elegans* and *Homo sapiens* UCP4 orthologues had been functionally linked to a succinate transport in mitochondria [16,17], we have also explored the possibility that *Dm*UCP4A could be a transporter of metabolites, as UCP2, UCP5, and UCP6 in mammals, as well as UCP1 and UCP2 in plants [4,5,6].

Here, we show that the recombinant *Dm*UCP4A protein reconstituted into liposomes or expressed in *Saccharomyces cerevisiae* mitochondria catalyzes an efficient unidirectional transport of aspartate. A possible physiological role of *Dm*UCP4A in fruit fly metabolism has been discussed.

## 2. Results

### 2.1. Bacterial Expression of DmUCP4A and Optimization of the Transport Assay into Liposomes

The *Dm*UCP4A protein was expressed at high levels in the *E. coli* BL21-CodonPlus (DE3)-RIL heterologous system. It accumulated in the bacterial cytoplasm as inclusion bodies (IB) (Figure 1A, lanes 4) and no protein was detected in bacteria carrying the *Dm*UCP4A expression vector harvested before the induction of expression (Figure 1A, lane 2). IB were purified on a sucrose gradient followed by Triton X114 washing (Figure 1A, lane 5). The identity of the purified protein was confirmed by Western blot analysis (Figure 1B lanes 4 and 5). Approximately 20–30 mg of purified protein per liter of culture was obtained. 

No protein was detected in bacteria transformed with the empty vector and harvested before and after the induction of expression (Figure 1A,B, lanes 1 and 3).

The sarkosyl-solubilized *Dm*UCP4A IB were initially reconstituted into liposomes by following a standard reconstitution protocol, which had been successfully used to characterize many MCF members [5,18,19,20]. A variety of substrates, nucleotides, amino acids, and organic acids were assayed in homo-exchange experiments (i.e., using the same substrate inside and outside the proteoliposomes). Using external (radioactive) and internal (unlabeled) substrate concentrations of 1 and 10 mM, respectively, *Dm*UCP4A catalyzed only an [^14^C]aspartate/aspartate exchange reaction (Figure 1C), which was completely inhibited by a mixture of pyridoxal-5′-phosphate (PLP) and bathophenanthroline (BAT), two well-known inhibitors of other MCF members [4,20,21]. Virtually no homo-exchange activity was measured using ATP, ADP, succinate, malate, 2-oxoglutarate, malonate, glutamate, and γ-aminobutyric acid (Figure 1C). Similarly, no detectable [^14^C]aspartate/aspartate exchange activity was found using boiled, unfolded reconstituted *Dm*UCP4A (data not shown).

In order to find the best reconstitution conditions to assay *Dm*UCP4A in vitro, different amounts of sarkosyl-solubilized *Dm*UCP4A and activating lipids including cardiolipin (DPG) [19,20] and asolectin (ASO) [22,23] were tested in the reconstitution mixture. The highest [^14^C]aspartate/aspartate exchange activity was found by reconstituting an amount of 10 μg of protein in the presence of 0.6–0.8 mg/mL DPG (Figure 2A,B).

*Dm*UCP4A showed a sharp sensitivity to the reconstitution mixture pH value, with an optimum between 6 and 6.5 (Figure 2C). On the basis of these initial results, the following experiments were carried out using 10 μg of solubilized *Dm*UCP4A, 0.6 mg/mL DPG at pH 6.5.

### 2.2. Kinetic Properties of the Recombinant DmUCP4A Protein

The uptake of 0.5 mM [^14^C]aspartate into proteoliposomes was measured either as exchange (in the presence of internal 10 mM aspartate) or uniport (in the absence of an internal substrate). In preliminary experiments, *Dm*UCP4A seemed to catalyze both exchange and uniport reactions. A possible unidirectional transport could invalidate some of our transport results because, during the removal of the external unlabeled substrate by gel exclusion chromatography (see Section 4), a leak of the internal substrate (unidirectionally transported) would have changed the concentration of unlabeled substrate inside the proteoliposomes. Since the *Dm*UCP4A protein sequence contains seven cysteines, we tried to avoid this technical inconvenience by using the reversible inhibitor p-chloromercuribenzenesulfonate (pCMBS), able to bind to thiol groups, before loading the proteoliposomes on chromatographic columns. Then, the labeled substrate was externally added together with dithioerythritol (DTE), which was able to bring back the protein to an active state [19] (see Section 4).

As expected, *Dm*UCP4A catalyzed both exchange and uniport reactions following first-order kinetics, with isotopic equilibrium being approached exponentially (Figure 3A). The ratio of maximal substrate uptake by exchange and uniport was 21.2, in good agreement with the expected value of 20 from intraliposomal concentrations at equilibrium (0.5 and 10 mM for uniport and exchange, respectively). The initial rates of aspartate uptake deduced from the time courses were 1.48 and 0.066 μmol/min/mg of protein for exchange and the uniport reaction, respectively. 

The unidirectional transport catalyzed by *Dm*UCP4A was further investigated by measuring the efflux of [^14^C]aspartate from preloaded active proteoliposomes (Figure 3B) because this experimental approach has the greatest sensitivity for unidirectional transport [24,25]. Using the reconstituted *Dm*UCP4A protein, a substantial and rapid efflux occurred upon the addition of DTE together with unlabeled aspartate (●) or, to a lower extent, upon the addition of DTE plus piperazine-N,N′-bis(2-ethanesulfonic acid) (PIPES), pH 6.5, which does not act as a substrate (▲) (Figure 3B). These results have demonstrated that aspartate transport can also occur unidirectionally because in the absence of an external counterion a significant efflux of a radioactive substrate from proteoliposomes has been observed. Furthermore, the exchange reaction was completely prevented by adding inhibitors, PLP and BAT (■) (Figure 3B). The unidirectional efflux of labeled aspartate from proteoliposomes did not significantly change if a 1.5 unit of ΔpH (▼) (6.5 inside and 8 outside) was applied across the proteoliposomal membrane, thus suggesting that *Dm*UCP4A is able to catalyze an electrogenic rather than a proton-assisted uniport of aspartate (Figure 3B). We were unable to dissect this point further by generating a valinomycin-dependent K^+^-diffusion potential across the proteoliposomal membrane [6,26], since the high concentrations of KCl required in these kinds of experiments interfered with *Dm*UCP4A transport activity.

The kinetic constants of the recombinant purified *Dm*UCP4A protein were determined by measuring the initial transport rate at various external [^14^C]aspartate concentrations, in the presence of a constant saturating internal aspartate concentration (10 mM). The *K_m_* values of the recombinant *Dm*UCP4A protein for aspartate was 229.2 ± 12.64 µM. In five experiments, the *Dm*UCP4A *V*_max_ value, corrected for small differences in the efficiency of reconstitution, was 2.491 ± 0.215 µmol/min/mg of protein.

### 2.3. Substrate Specificity and Inhibitor Sensitivity of the Recombinant DmUCP4A Protein

Substrate specificity of the purified recombinant *Dm*UCP4A protein was examined in detail by measuring the uptake of [^14^C]aspartate into proteoliposomes preloaded with various unlabeled substrates. *Dm*UCP4A efficiently transported only aspartate among all tested substrates. A very low exchange activity was found with oxalate, maleate, cysteine sulfinate, 2-oxoglutarate, and thiosulfate, whereas the exchange activities of the remaining tested substrates were approximately the same as that observed in the absence of an internal substrate (NaCl) (Figure 4A). No significant exchange activities were also found in proteoliposomes preloaded with di- and triphosphate nucleotides (not shown).

In order to confirm the strong aspartate specificity of *Dm*UCP4A, some substrates used in the previous experiment were externally used as potential competitive inhibitors of the [^14^C]aspartate/aspartate exchange reaction (Figure 4B). All substrates were added at a concentration about thirty-fold higher than that of labeled aspartate (i.e., 10 mM vs. 0.3 mM). In this experiment, unlabeled aspartate was also used as a positive control. The exchange reaction catalyzed by *Dm*UCP4A was strongly inhibited by the external addition of aspartate, and to a much lower extent by oxalate, maleate, 2-oxoglutarate, and cysteine sulfinate, no significant inhibition was observed with the other tested substrates (Figure 4B).

The effect of known mitochondrial transporter inhibitors [5,20] on the aspartate/aspartate exchange reaction catalyzed by the reconstituted recombinant *Dm*UCP4A protein was also examined (Figure 4B). The activity of *Dm*UCP4A was strongly inhibited by BAT, PLP, and tannic acid. As expected, thiol reagents such as mersalyl, pCMBS, and pHMB were powerful inhibitors of *Dm*UCP4A transport function. The inhibition experiments with thiol reagents were carried out without the addition of the reversible inhibitor pCMBS. Although *Dm*UCP4A did not transport dicarboxylates, butylmalonate, and phenylsuccinate, known inhibitors of the dicarboxylate and 2-oxoglutarate carriers [18,27,28] strongly inhibited the [^14^C]aspartate/aspartate exchange reaction (Figure 4B). 

### 2.4. DmUCP4A Catalyzes a Unidirectional Transport of Aspartate in Saccharomyces Cerevisiae Mitochondria

The unidirectional aspartate transport catalyzed by *Dm*UCP4A was further investigated in a more physiological condition, i.e., when the protein was expressed in the yeast mitochondrial membrane. The yeast mitochondrial aspartate/glutamate transporter (AGC1p) catalyzes an aspartate/glutamate exchange reaction and a unidirectional proton-assisted transport of both amino acids as well [29]. In yeast mitochondria, a proton-assisted unidirectional transport of an anion can be assayed by swelling experiments carried out in an isosmotic solution containing the ammonium salt of the tested anion [4,30]. Mitochondrial swelling occurs when large amounts of a solute enter the matrix space, causing increased osmotic pressure and volume. Swelling occurs only if the movement of the solute is charge-balanced; this happens when the anion is transported together with a proton. Mitochondrial swelling also occurs if the anion is unidirectionally transported without a proton. In this case, swelling occurs in a potassium salt of the anion upon the addition of valinomycin, which counterbalances the negative charge of the anion by transporting potassium. In our experiments, wild-type yeast mitochondria did not swell in ammonium chloride (negative control), but they swelled in ammonium phosphate due to the presence of the mitochondrial phosphate carrier (PiC), which catalyzes a Pi/H^+^ symport (positive control) (Figure 5A). 

As expected, yeast mitochondria also swelled in the presence of ammonium salt of aspartate, but not in that of potassium, even upon the addition of valinomycin (Figure 5A), confirming that yeast AGC1p is able to catalyze an aspartate/H^+^ symport.

Thus, an AGC1p deleted yeast strain (*agc1*Δ), expressing *Dm*UCP4A (Figure 6E), was used to dissect the unidirectional transport catalyzed by the fruit fly transporter. AGC1p deleted mitochondria swelled in ammonium phosphate, but did not swell in ammonium aspartate (Figure 5B). Similar results were obtained with mitochondria expressing *Dm*UCP4A (Figure 5C), suggesting that the unidirectional transport of aspartate found in liposomes could not be proton-assisted. This hypothesis was further confirmed by the swelling observed in mitochondria expressing *Dm*UCP4A in the presence of potassium aspartate upon the addition of valinomycin (Figure 5C), but not in those of the AGC1p deleted strain. Both types of mitochondria did not swell in potassium salts of succinate and malate (Figure 5B,C). Thus, swelling experiments confirm that the unidirectional aspartate transport catalyzed by *Dm*UCP4A is electrogenic and not proton-coupled.

### 2.5. DmUCP4A Catalyzes a Unidirectional Transport of Aspartate in a Yeast Cell Model

Our swelling experiments excluded any possible involvement of protons in the unidirectional transport of aspartate catalyzed by *Dm*UCP4A. Being the electrical gradient across the inner mitochondrial membrane positive outside, and aspartate transported with a net negative charge, *Dm*UCP4A could be involved in the export of aspartate out of mitochondria. This hypothesis was further tested and verified in a yeast cell model. Yeast cells use the malate-aspartate shuttle (MAS) to reduce the cytosolic NADH derived from peroxisomal oleate oxidation [29]. AGC1p is an essential component of MAS, hence its deletion makes yeast cells unable to grow on oleate [29]. Since another member of the MCF, YMC2p, is present in yeast mitochondria and is able to catalyze a Glu/H^+^ symport [31], we hypothesized that if *Dm*UCP4A catalyzed the export of aspartate from mitochondria and operated together with YMC2p, then aspartate/glutamate exchange activity would be re-established and the growth defect on oleate of the AGC1p-deleted yeast strain could be rescued.

The *agc1*Δ yeast strain did not show any growth defect on glucose (not shown) [29], but it did not grow on oleate (Figure 6A,C). The expression of *Dm*UCP4A or the re-introduction of the endogenous transporter in the *agc1*Δ yeast strain rescued the growth defect on oleate. In order to further confirm that the rescued phenotype was due to *Dm*UCP4A transport activity and that of the endogenous yeast mitochondrial glutamate/H^+^ symporter (YMC2p), we constructed a double mutant strain lacking AGC1 and YMC2 (*agc1*Δ*/ymc2*Δ). As shown in Figure 6B,D, the expression of *Dm*UCP4A was unable to complement the growth defect of the double mutant. Hence, our results confirm that the unidirectional transport of aspartate catalyzed by *Dm*UCP4A occurs not only in vitro but also in a cellular context.

## 3. Discussion

Mitochondrial UCPs belong to the mitochondrial carrier family, a group of proteins mainly localized in the inner mitochondrial membrane, which shuttle metabolites and cofactors across this membrane by connecting cytosolic metabolic pathways to those occurring in the mitochondrial matrix. Although for many years the physiological role of all UCPs has been associated with their fatty acids-dependent protonophoric activity, this biochemical function has been unequivocally demonstrated only for the UCP1 archetypal uncoupler [3,32]. Furthermore, many members of the UCP subfamily of different species have been shown to transport metabolites [4,5,6,33] as other members of the MCF [2,34]. In the fruit fly, there are four members of the MCF, on the basis of their homology sequence with other UCPs, they have been classified as uncoupling proteins, *Dm*UCP4A-C and *Dm*UCP5. A protonophoric activity has been suggested for three of them, *Dm*UCP4B, *Dm*UCP4C, and *Dm*UCP5 [12,13,14]. No functional data are available on *Dm*UCP4A, although its involvement in any uncoupling activity has been ruled out [10,13]. Here, we demonstrate that *Dm*UCP4A unidirectionally transports only aspartate, both in vitro studies and using a yeast cell model. 

Of note, the biochemical function of UCP4 orthologs in neuroblastoma cells and *C. elegans* (*Ce*UCP4) has been linked to the activity of complex II [16,17,35]. Transport studies carried out on intact nematode mitochondria have suggested that *Ce*UCP4 could be involved in the transport of succinate [16]. In our reconstituted system, as well as in yeast mitochondria, the recombinant *Dm*UCP4A protein does not transport succinate or other dicarboxylates (Figure 4A and Figure 5C). It should be considered that a possible aspartate transport catalyzed by *Ce*UCP4 has been not assayed [16]. Furthermore, only one UCP homologous transporter has been found in this nematode, suggesting that it might have a different biochemical/physiological role. A similar consideration can be made for *Dm*UCP4A compared to mammalian UCPs. The fruit fly genome does not contain any gene orthologous to the human UCP1-3 [7], thus we cannot exclude that the biochemical function of *Dm*UCP4A, as well as that of *Dm*UCP4B, *Dm*UCP4C, and *Dm*UCP5, might overlap that of mammalian UCPs other than UCP4 and UCP5. In fact, *Dm*UCP4A transports aspartate as the human UCP2 [4], although their substrate specificity and transport mechanism are significantly different: (i) UCP2 transports aspartate, malate, oxaloacetate, malonate, phosphate, and sulfate, whereas *Dm*UCP4A transports only aspartate (Figure 1C, Figure 4A and Figure 5C); (ii) similarly to UCP2, *Dm*UCP4A catalyzes a unidirectional transport of substrate, but this latter is not proton-coupled, conversely, UCP2 catalyzes a Pi/H^+^ symport [4]. The unidirectional transport of aspartate catalyzed by *Dm*UCP4A would be favored by the electrical gradient across the inner mitochondrial membrane. A discrepancy in substrate specificity or transport mechanism between *D. melanogaster* and the human orthologs is not surprising, since it has already been reported for other mitochondrial carriers [28,36]. In mammals, UCP2 plays a key role in transporting the glutamine-derived aspartate from mitochondria to the cytosol, in which it is used for nucleotide and protein synthesis, as well as in redox homeostasis [4,33,37,38]. 

In *D. melanogaster*, aspartate plays a key role not only in protein and pyrimidine synthesis [39,40] but also in the synthesis of β-alanine [41]. Unlike mammals, β-alanine is a crucial substrate for histamine recycling in the visual system [42,43], for dopamine recycling in the central nervous system, and for cuticle pigmentation [44,45]. 

In detail, β-alanine can be synthesized by the decarboxylation of aspartate catalyzed by glutamate decarboxylase isoform 2 [46]. In the fruit fly nervous system, β-alanine is a key player in the recycling of biogenic amines histamine and dopamine [47]. Although histamine recycling is well known [41,48], few data are available for that of dopamine, even if both mechanisms are supposed to act similarly to the known mammalian tripartite synaptic cycle [49]. Biogenic amine recycling consists of three elements: a glial cell, a presynaptic, and a postsynaptic neuron [47]. In glial cells, N-β-alanyl dopamine synthase (encoded by the *ebony* gene) catalyzes the condensation of β-alanine with histamine or dopamine in order to form β-alanyl-histamine, also known as carcinine [48], or N-β-alanyl-dopamine, respectively. The synthesized substrates are transported to presynaptic neurons, in which N-β-alanyl dopamine hydrolase (encoded by the *tan* gene) cleaves them into the original neurotransmitters, which are secreted into postsynaptic neurons where they explicate their function [47]. The interruption of these recycling systems affects the normal functioning of the visual system [48], circadian activity rhythms [50], as well as cuticle formation and pigmentation [43].

Furthermore, *Dm*UCP4A may be involved in N-acetylaspartate metabolism. In humans, the cytosolic availability of aspartate is crucial for the synthesis of N-acetylaspartate, a precursor of myelin biosynthesis [51,52]. Interestingly, in PD, a decrease in N-acetylaspartate level has been found [53,54,55,56,57,58], and the stimulation of the dopamine D1 receptor through an agonist could increase the expression of Nat8L, i.e., the enzyme responsible for N-acetylaspartate biosynthesis in mammals. Mammalian Nat8L has been found in mitochondrial and microsomal fractions, although the latter contains the most active enzyme [59,60]. Thus, a mitochondrial exporter of aspartate should increase the cytosolic availability of aspartate required for N-acetylaspartate biosynthesis. Although no data are available on this biosynthetic pathway in the fruit fly, its existence has recently been proven [61]. *Dm*UCP4A could improve the phenotype of fruit fly models of Parkinson’s disease [10,11] by increasing the cytosolic availability of aspartate for N-acetylaspartate biosynthesis. Our results clearly demonstrate that *Dm*UCP4 is involved in the export of aspartate from mitochondria, although further studies in *D. melanogaster* or cell models are required to establish the specific metabolic pathways in which the aspartate transport function of *Dm*UCP4 could be required.

## 4. Materials and Methods

### 4.1. Construction of Plasmids for D. melanogaster DmUCP4A Expression in E. coli and S. cerevisiae

*Dm*UCP4A was amplified by polymerase chain reaction (PCR) using the fruit fly cDNA clone RH64870 as a template (*Drosophila* Genomics Resource Center, Indiana University 1001 East Third St., Bloomington, IN, USA, 47405-7107). The oligonucleotide primers (forward and reverse) used in PCR reactions carried appropriate restriction sites at their 5′ ends for the further cloning of the amplified inserts in the pET-21b/V5-His *E. coli* expression vector [62,63] and in the pYES2/V5-His yeast expression vector, in which the inducible GAL1 promoter had been replaced by the constitutive TDH3 promoter [33]. The absence of a stop codon in the reverse primer sequence led to the expression of the *Dm*UCP4A protein containing a V5/His-tag at its C-terminus [64]. The endogenously deleted yeast genes AGC1 and YMC2 were cloned in the centromeric pRS416 and pRS413 vectors, respectively, and used to verify the growth defect phenotype in oleate.

### 4.2. Bacterial Expression and Purification of DmUCP4A

The overexpression of V5-tagged *Dm*UCP4A resulted in the production of IB in the cytosol of *E. coli* BL21-CodonPlus (DE3)-RIL, and it was accomplished as previously described [65]. IB were purified on a sucrose density gradient and washed at 4 °C, firstly with TE buffer (10 mM Tris⁄HCl, 1 mM ethylenediaminetetraacetic acid (EDTA), pH 8), then twice with a buffer containing Triton X-114 (2%, *w*/*v*) and 10 mM PIPES (pH 6.5), and finally with PIPES 10 mM (pH 6.5). The identity of the recombinant protein was assayed by immunoblotting [66] using an anti-V5 specific antiserum (Cell Signaling). The recombinant *Dm*UCP4A protein was solubilized in 1.8% (*w*/*v*) sarkosyl.

### 4.3. Reconstitution into Liposomes and Transport Assays

The recombinant protein solubilized in sarkosyl was reconstituted into liposomes in the presence or absence of substrates [67]. The reconstitution mixture contained the sarkosyl-solubilized *Dm*UCP4A protein (10 μg of protein), 1% (*w*/*v*) Triton X114, 1.3% (*w*/*v*) egg yolk phosphatidylcholine (Sigma-Aldrich, St. Louis, MO, USA) sonicated liposomes, 0.6 mg/mL cardiolipin (Sigma-Aldrich), 10 mM unlabeled substrate (except where otherwise indicated), 10 mM PIPES at pH 6.5 (except where otherwise indicated), and water to a final volume of 700 μL. The mixture was vortexed and recycled 13 times through the same Amberlite column (Bio-Rad) [20].

External unlabeled substrate was removed from proteoliposomes on a Sephadex G-75 column pre-equilibrated with 50 mM NaCl and 10 mM PIPES (pH 6.5). The pCMBS (10 μM) was added to proteoliposomes before loading on Sephadex G-75, in order to avoid leakage of an internal substrate during the chromatographic process. Transport at 25 °C was started by adding the labeled substrate, at the indicated concentrations, to unlabeled substrate-loaded proteoliposomes (exchange) or to empty proteoliposomes (uniport). The inhibition of transport exerted by pCMBS was removed by adding 5 mM DTE together with the labeled substrate [19]. The transport was terminated, at the desired time point, by the addition of 10 mM PLP and 10 mM BAT. In control samples, these inhibitors were added at time 0, according to the inhibitor stop method [68].

Finally, the external substrate was removed and radioactivity into proteoliposomes was measured. Experimental values were corrected by subtracting control values. The initial transport rate was calculated from the radioactivity taken up by proteoliposomes after 1 min (in the initial linear range of substrate uptake). For efflux measurements, proteoliposomes containing 5 mM aspartate were labeled with 20 µM L-[^14^C]aspartate by carrier-mediated exchange equilibration. After 60 min, the external substrate was removed by exclusion chromatography in the presence of a reversible inhibitor (10 μM pCMBS). The Sephadex G-75 columns used in this kind of experiment had been pre-equilibrated with 50 mM NaCl and 0.1 mM PIPES, pH 6.5. The reduced PIPES concentration allowed to change the pH value during the transport reaction. Efflux was started by adding 20 mM aspartate plus 5 mM DTE (exchange reaction) or PIPES 20 mM (pH 6.5 or 8) with 5 mM DTE (uniport reaction). In all cases, the transport was terminated by adding PLP (10 mM) and BAT (10 mM). 

### 4.4. Yeast Strains, Growth Conditions, and Functional Complementation

A W303 (wild-type) (*MATa {leu2*–*3112 trp1*–*1 can1*–*100 ura3*–*1 ade2*–*1 his3*–*11,15}*) yeast strain was provided by the EUROFAN resource center EUROSCARF (Frankfurt, Germany). The yeast gene deletions (*agc*1Δ and *agc*1Δ/*ymc2*Δ) were achieved as described before [26,33]. The recombinant *Dm*UCP4A-pYES2_TD3H [33] and pRS416 plasmids, without or with the coding sequence of AGC1p, respectively, were introduced into the *agc1*Δ yeast strain. The recombinant *Dm*UCP4A- pYES2_TD3H and pRS413 plasmids, without or with the coding sequence of YMC2p, respectively, were introduced into the *agc1*Δ*/ymc2*Δ yeast strain. Yeast cells were transformed using the lithium acetate method [69], and transformants were selected on synthetic complete medium plates lacking uracil (*agc1*Δ) or uracil and histidine (*agc1*Δ*/ymc2*Δ). Functional complementation was carried out by growing cells on liquid complete medium (1% Bacto yeast extract, 2% Bacto Peptone, (YP) pH 5.0) supplemented with 2% glucose (YPD) or 0.5 mM oleic acid dissolved in 10% Tween 40 (YPO) as carbon sources. Growths were started from medium log precultures grown on complete medium YPD and diluted with YPO to an optical density of 0.05 at 600 nm. Simultaneously, washed cells were diluted and spotted on complete solid medium YPO, then plates were incubated for 72 h at 30 °C. Four-fold serial dilutions of both the wild-type, deleted, and transformed strains were analyzed.

### 4.5. Isolation and Swelling of Yeast Mitochondria

Mitochondria were isolated by standard procedures from the wild-type, *agc1*Δ, and *agc1*Δ transformed with *Dm*UCP4A-pYES2_TD3H yeast strains grown in YP medium containing 2% ethanol until the early exponential phase (optical density between 1.0 and 1.5) was reached [70,71]. The rate of mitochondrial swelling was measured by recording the decrease in A_546_ using a Jenway 7315 spectrophotometer, as previously described [72]. Yeast mitochondria (500 µg of protein) were added to a glass cuvette containing 1 mL 120 mM NH_4_Cl and NH_4_Pi or 80 mM ammonium and potassium salts of aspartate, succinate and malate, 20 mM Tris, 1 mM EDTA, 5 µM rotenone, and 0.1 µM antimycin, pH 7.4. 

### 4.6. Other Methods

Proteins were resolved by SDS-PAGE and stained with Coomassie blue dye or transferred onto a nitrocellulose membrane by a Trans-Blot Turbo Transfer System (Bio-Rad Laboratories). Western blotting was carried out with a mouse anti-V5 monoclonal antibody (Cell Signaling) [66,73]. A rabbit antiserum against yeast porin [74], a kind gift from L. Pelosi, was used for protein normalization. Immunoreactive proteins were detected by enhanced chemiluminescence. Moreover, the amounts of recombinant *Dm*UCP4A incorporated into liposomes were measured by laser densitometry of stained samples, as previously described [75], and proved to be approximately 20% of the protein amount added to the reconstitution mixture.

## Figures and Tables

**Figure 1 ijms-23-01020-f001:**
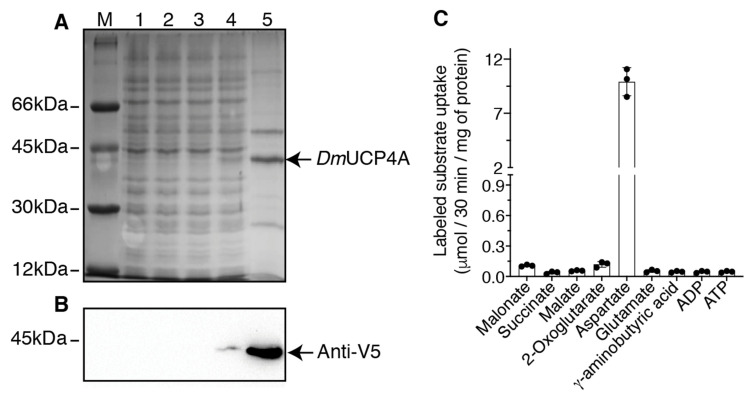
Overexpression in *E. coli* and partial purification of *Dm*UCP4A and its initial reconstitution into liposomes. (**A**) Proteins were separated by SDS-PAGE and stained with Coomassie Blue. Lane M, markers (bovine serum albumin (66 kDa), ovalbumin (45 kDa), carbonic anhydrase (30 kDa), and cytochrome c (12 kDa), lanes 1–4, *E. coli* BL21-CodonPlus (DE3)-RIL containing the expression vector with the coding sequence of *Dm*UCP4A carrying a V5-tag at the C-terminus, (lanes 2 and 4), or the empty vector (lanes 1 and 3)). Samples were taken at the time of induction (lanes 1 and 2) and 5 h later (lanes 3 and 4). The same number of bacteria was analyzed in each sample. Lane 5, *Dm*UCP4A inclusion bodies (IB) originating from bacteria shown in lane 4. (**B**) Immunoblotting of the samples reported in (**A**) carried out with a specific anti-V5-tag antibody. (**C**) An amount of 20 μg of sarkosyl-solubilized IB was reconstituted in the presence of 1% (*w*/*v*) Triton X114, various unlabeled substrates (each at a final concentration of 10 mM), 1.3% (*w*/*v*) egg yolk phosphatidylcholine and 0.5 mg/mL cardiolipin, at pH 7. The exchange reaction was started by the addition of various labeled substrates (each at a final concentration of 1 mM) and terminated after 30 min. The means of two technical replicates of three independent experiments were reported.

**Figure 2 ijms-23-01020-f002:**
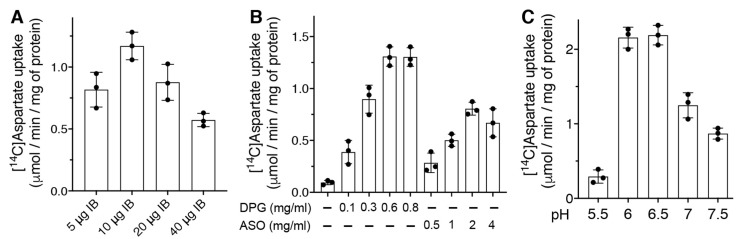
Optimization of the reconstituted recombinant *Dm*UCP4A transport activity into liposomes. (**A**) Different amounts of sarkosyl-solubilized *Dm*UCP4A IB were reconstituted in the presence of 1% (*w*/*v*) Triton X114, 10 mM aspartate, 1.3% (*w*/*v*) egg yolk phosphatidylcholine and 0.5 mg/mL cardiolipin (DPG), at pH 7. The exchange reaction was started by the addition of 1 mM [^14^C]aspartate and terminated after 1 min. (**B**) An amount of 10 μg of sarkosyl-solubilized *Dm*UCP4A IB was reconstituted, as reported in (**A**), in the presence of different amounts of DPG or asolectin (ASO). (**C**) An amount of 10 μg of sarkosyl-solubilized *Dm*UCP4A was reconstituted, as reported in (**A**), in the presence of 0.6 mg/mL DPG at various pH values. The means of two technical replicates of three independent experiments were reported.

**Figure 3 ijms-23-01020-f003:**
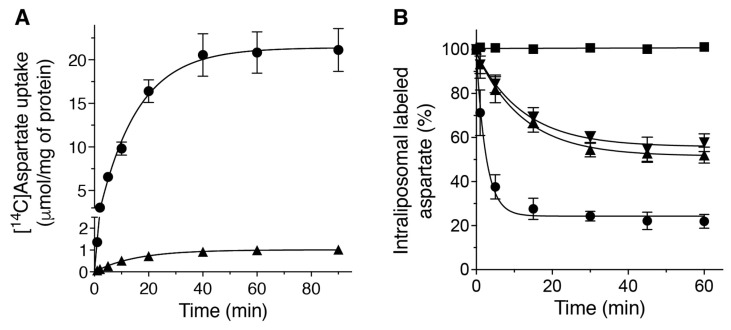
Kinetics of [^14^C]aspartate uniport and [^14^C]aspartate/aspartate exchange reactions catalyzed by *Dm*UCP4A. Proteoliposomes were reconstituted with *Dm*UCP4A and assayed in forward (**A**) and backward (**B**) modes, respectively (see Section 4 for details). (**A**) A concentration of 0.5 mM [^14^C]aspartate was added to proteoliposomes containing 10 mM aspartate (exchange, ●) or 10 mM NaCl in the absence of substrate (uniport, ▲). Transport was terminated by the addition of inhibitors such as 10 mM pyridoxal-5′-phosphate (PLP) and 10 mM bathophenanthroline (BAT) at the desired time point. (**B**) A concentration of 5 mM intraliposomal aspartate was labeled with [^14^C]aspartate by carrier-mediated exchange equilibration. After removing the external substrate by Sephadex G-75 (pre-equilibrated with 50 mM NaCl and 0.1 mM piperazine-N,N′-bis(2-ethanesulfonic acid) (PIPES), pH 6.5) and in the presence of the reversible inhibitor p-chloromercuribenzenesulfonate (pCMBS), the efflux of [^14^C]aspartate was started by adding a buffer containing 20 mM aspartate plus 5 mM dithioerythritol (DTE) (exchange, ●), 20 mM aspartate plus 5 mM DTE and inhibitors (■), 20 mM PIPES, pH 6.5, plus 5 mM DTE (uniport, ▲) or 20 mM PIPES, pH 8, plus 5 mM DTE (uniport, 1.5 unit of ΔpH, ▼) and terminated by the addition of inhibitors at the desired time point. The means of two technical replicates of three independent experiments were reported.

**Figure 4 ijms-23-01020-f004:**
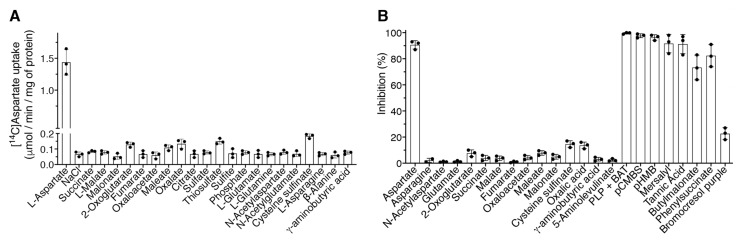
Substrate specificity and inhibitor sensitivity of the reconstituted recombinant *Dm*UCP4A protein. (**A**) Proteoliposomes were internally preloaded with various substrates (10 mM). Transport was started by the addition of 0.3 mM [^14^C]aspartate. The reaction time was 1 min. The means of two technical replicates of three independent experiments were reported. (**B**) Effect of externally added unlabeled substrate on the [^14^C]aspartate/aspartate exchange reaction catalyzed by *Dm*UCP4A and inhibitor sensitivity. Proteoliposomes were internally preloaded with 10 mM aspartate. Transport was initiated by adding 0.3 mM [^14^C]aspartate to proteoliposomes reconstituted with *Dm*UCP4A. All externally unlabeled substrates (used at 10 mM) were added together with the labeled substrate, thiol reagents were added 2 min before the labeled substrate. The final concentrations of the inhibitors were 10 mM BAT and PLP, 10 µM mersalyl, pCMBS, and p-hydroxymercuribenzoate (pHMB); 0.2% tannic acid (TAN); 10 mM butylmalonate and phenylsuccinate, and 0.3 mM bromcresol purple. The reaction time was 1 min. The means of the extent of inhibition (%) of two technical replicates of three independent experiments were reported.

**Figure 5 ijms-23-01020-f005:**
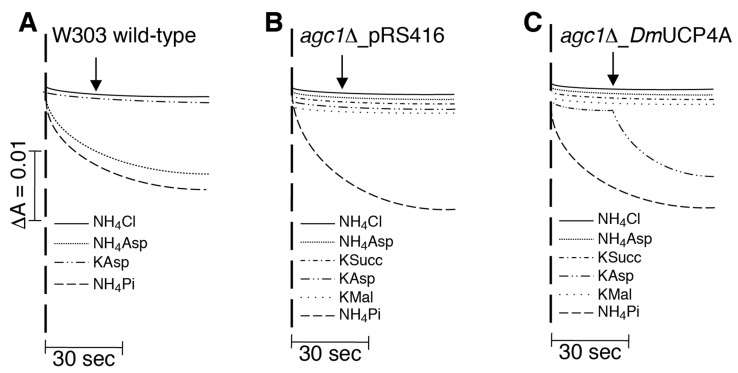
Swelling of wild-type W303, *agc1*Δ, and *agc1*Δ expressing *Dm*UCP4A yeast mitochondria in isosmotic ammonium or potassium solutions of various anions. (**A**–**C**) Mitochondria (0.1 mg of protein) were suspended at 25 °C in a solution containing ammonium or potassium salts (120 mM NH_4_Cl and NH_4_Pi; 80 mM ammonium and potassium salts of aspartate (Asp), succinate (Succ), and malate (Mal)) of the indicated anions, 20 mM Tris, pH 7.4, 1 mM (ethylenediaminetetraacetic acid) EDTA, 0.1 mM antimycin, and 5 mM rotenone in a final volume of 1 mL. Turbidity changes of the mitochondrial suspensions were recorded at 546 nm. Arrows indicate the addition of 10 μM valinomycin (see text for more details).

**Figure 6 ijms-23-01020-f006:**
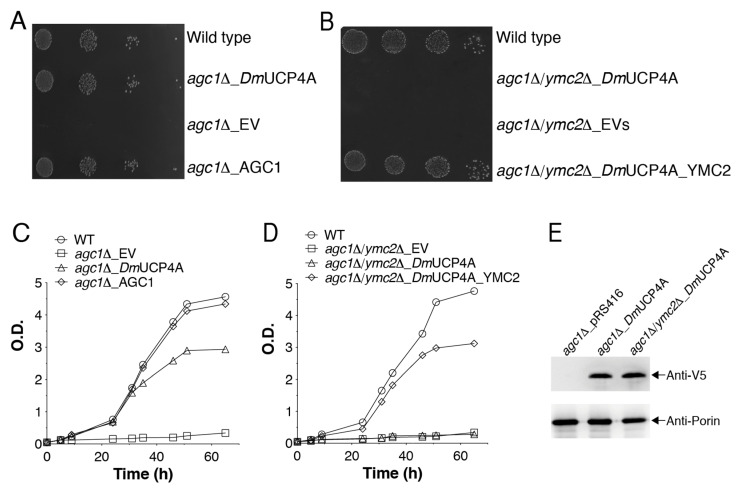
*Dm*UCP4A functions as an aspartate uniporter in yeast cells. (**A**,**C**) Growth behavior of *S. cerevisiae* W303 *agc1*Δ and (**B**,**D**) *agc1*Δ/*ymc2*Δ cells expressing *Dm*UCP4A grown on solid (**A**,**B**) or liquid medium (**C**,**D**) containing oleate as a sole carbon source. (**A**,**B**) Various transformed yeast strains were normalized in water to an optical density of 0.4 at 600 nm. Four serial dilutions of the different cell models were spotted onto YP plates in the presence of 0.5 mM oleate. Plates were placed at 30 °C, and pictures were taken after 3 days to show yeast growth performance. (**C**,**D**) Various transformed yeast strains were inoculated, starting from the same optical density, in YP medium in the presence of 0.5 mM oleate. The values of optical density at 600 nm refer to cell cultures after the indicated times of growth. (**E**) A representative immunoblot of the expression of the *Dm*UCP4AV5-tagged protein in *agc1*Δ and *agc1*Δ/*ymc2*Δ yeast strains. A yeast anti-porin antiserum was used for protein normalization.

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
