# Peer review of "Drosophila melanogaster Uncoupling Protein-4A (UCP4A) Catalyzes a Unidirectional Transport of Aspartate"

_ijms, 2022, doi:10.3390/ijms23031020_

Round 1

Reviewer 1 Report

Overall the paper is good and well written.

The author has to change the organism name in italic.

eg.

specie to be biochemically characterized. In Drosophila melanogaster, DmUCP4A could be in- 37

a-helices, four Drosophila melanogaster MCF members were classified as uncoupling pro-

Kindly check the degree sign in all occurrences.

Please check the y axis of Figure 2C. 

kindly check the manuscript for typo errors.

Author Response

First of all, many thanks for the positive comments on the manuscript

The author has to change the organism name in italic.

eg.

specie to be biochemically characterized. In Drosophila melanogaster, DmUCP4A could be in- 37

a-helices, four Drosophila melanogaster MCF members were classified as uncoupling pro-

Reply: The whole text has been checked and all Drosophila, Dm, D. melanogaster and Drosophila melanogaster have been reported in italics.

Kindly check the degree sign in all occurrences.

Reply: It has been done.

Please check the y axis of Figure 2C.

Reply: The y axis labelling of Figure 2C has been corrected. 

kindly check the manuscript for typo errors.

Reply: It has been done.

Reviewer 2 Report

I consider the detailed biochemical characterization of this drosophila UPC transporter (UPC4A) as a significant study since earlier the role of UPC4A was interpreted mainly on hypothetical functions, although it rescued PD model mutants of Drosophila in several points. The experiments are convincing, I have only minor formal comments:

Fig 3. - the graph curve labels are missing in the figure text (or may be it lost 1. when pdf was formatted) whereas some of them appeares in the main text (row 204)

2. Row 206, 210, 213 There are letters in brackets (B), (P) that I can not identify - probably missing labels of graph curves?

3. Row 223. The unit would be correctly umol/min/mg or use bracket

4. Regarding the discussion. There is a lengthy discussion of Parkinson disease and potential connections of it with the revealed function of the investigated transporter, and at present this is more than uncertain. But I found the discussion carefully phrased and no need of omitting this part. I agree with last sentence that further studies are needed to reveal the significance of aspartate transport by UCP4A in alanine and N-acetyl aspartate metabolism, before suggesting a role. 

Author Response

First of all, many thanks for the positive comments on the manuscript

Fig 3. - the graph curve labels are missing in the figure text (or may be it lost 1. when pdf was formatted) whereas some of them appeares in the main text (row 204)

Reply: We are sorry for that, during text formatting probably we used a symbol font that generate this kind of problem. We tried to fix it, please have a look to the PDF file of the revised manuscript if the problem persists.

  1. Row 206, 210, 213 There are letters in brackets (B), (P) that I can not identify - probably missing labels of graph curves?

Reply: Please see the previous answer

  1. Row 223. The unit would be correctly umol/min/mg or use bracket

Reply: the text has been changed in μmol/min/mg

  1. Regarding the discussion. There is a lengthy discussion of Parkinson disease and potential connections of it with the revealed function of the investigated transporter, and at present this is more than uncertain. But I found the discussion carefully phrased and no need of omitting this part. I agree with last sentence that further studies are needed to reveal the significance of aspartate transport by UCP4A in alanine and N-acetyl aspartate metabolism, before suggesting a role. 

Reply: Many thanks
